# Factors Contributing to Medication Adherence in Patients with a Chronic Condition: A Scoping Review of Qualitative Research

**DOI:** 10.3390/pharmaceutics13071100

**Published:** 2021-07-20

**Authors:** Kirsi Kvarnström, Aleksi Westerholm, Marja Airaksinen, Helena Liira

**Affiliations:** 1HUS Pharmacy, University of Helsinki and Helsinki University Hospital, 00290 Helsinki, Finland; 2Clinical Pharmacy Group, Division of Pharmacology and Pharmacotherapy, Faculty of Pharmacy, University of Helsinki, 00014 Helsinki, Finland; aleksi.westerholm@helsinki.fi (A.W.); marjaairaksinen@gmail.com (M.A.); 3Department of General Practice, University of Helsinki, 00290 Helsinki, Finland; helena.liira@helsinki.fi; 4Unit of Primary Health Care, University of Helsinki and Helsinki University Hospital, 00290 Helsinki, Finland

**Keywords:** medication adherence, patient compliance, primary non-adherence, patient-related factors, qualitative research, barriers, facilitators, scoping review, chronic conditions

## Abstract

Introduction: Medication adherence continues to be a significant challenge in healthcare, and there is a shortage of effective interventions in this area. This scoping review studied the patient-related factors of medication adherence. Methods: We searched Medline Ovid, Scopus, and Cochrane Library from January 2009 to June 2021 to find the most recent original qualitative studies or systematic reviews that addressed the patient-related factors of medication adherence in treating chronic conditions. We used the PRISMA-ScR checklist to ensure the quality of the study. Results: The initial search revealed 4404 studies, of which we included 89 qualitative studies in the scoping review. We inductively organized the patient-related factors causing barriers, as well as the facilitators to medication adherence. The studies more often dealt with barriers than facilitators. We classified the factors as patient-specific, illness-specific, medication-related, healthcare and system-related, sociocultural, as well as logistical and financial factors. Information and knowledge of diseases and their treatment, communication, trust in patient-provider relationships, support, and adequate resources appeared to be the critical facilitators in medication adherence from the patient perspective. Discussion and conclusions: Patients are willing to discuss their concerns about medications. Better communication and better information on medicines appear to be among the critical factors for patients. The findings of this scoping review may help those who plan further interventions to improve medication adherence.

## 1. Introduction

Medication adherence continues to be a significant challenge in healthcare, and there is a shortage of effective interventions. In 2003, the World Health Organization identified that only 50% of chronically ill patients take their medication as prescribed in developed countries [1]. Although there is a wealth of controlled trials on interventions to improve adherence, current methods of improving medication adherence for chronic health problems are mostly complex and not effective [2,3,4]. Previous studies and systematic reviews have combined the existing evidence of adherence interventions [3]. Nevertheless, it seems that there is still a lack of understanding about the complexity of medication adherence from the patient’s perspective.

Medication nonadherence is associated with poorer health outcomes [5]. If patients do not gain the expected health benefits from their medication because of nonadherence, the burden of health care costs increases for both patients and society in general [6,7,8]. The same factors that improve medication adherence may also decrease it [9,10,11]. The patient can experience medication-related burdens, which may negatively affect adherence [12]. On the other hand, a patient’s nonadherence can be seen as a behavioural problem related to their course of action [13]. Many studies have focused on medication adherence related to some specific illness instead of medication adherence in general. The studies may be lacking the input of patients, while the viewpoint of healthcare professionals may have been dominant. Patients struggle in reconciling daily life with comorbidity and multiple medications may be poorly understood [14]. Patient-centred care requires a greater understanding of the daily decisions patients need to make in order to manage a complex medication regimen.

Many theories have been applied to explain medication adherence behaviour. The information–motivation–behavioural skills (IMB) model is a widely used social behaviour model to explain medication adherence among chronically ill patients [15,16]. According to the model, the following three dimensions influence adherence behaviour: (1) Information and knowledge about the need for essential behaviour, (2) Motivation to make necessary behavioural changes and (3) The required behavioural skills to achieve the desired behaviour.

The model may explain patients’ actions regarding their rational use of medicines. Patients may not have sufficient information and understanding about their illness or medication to make an adequate decision, and they can seek the information from various sources [17]. Patient motivation is crucial to cope with multiple medications and make these fit into daily life. On the other hand, there may be personal reasons and system and organisation-specific barriers, which can lead to unwanted behaviour and medication nonadherence [18]. However, no theory alone seems to explain a patient’s adherence to medication because there may be external factors that can also affect adherence.

We need a more patient-focused approach to medication adherence and a better understanding of this complex phenomenon. This scoping review aimed at a better understanding of patients’ views on medication adherence and analysing the contributing factors as to why patients are not taking the medication as prescribed in outpatient settings. We wanted to understand this complex phenomenon in depth and summarize our findings in this scoping review.

## 2. Materials and Methods

We used the PRISMA-ScR checklist to ensure the quality of the study. The present scoping review is reported based on the guidelines of Preferred Reporting Items for Systematic reviews and Meta-Analyses extension for Scoping Reviews [19]. The PRISMA-ScR is available from the authors upon request.

### 2.1. Search Strategy

The literature search for eligible qualitative studies was conducted on 23 September 2019, using MEDLINE (Ovid), Scopus, and the Cochrane Library, with the assistance of an information specialist at the Helsinki University Library. The search was updated on 9 June 2021. We included articles that were published from January 2009 to June 2021. We wanted to focus on the most recent publications, so we did not include the publications published before the year 2009. We limited the article search to English language studies and articles published in peer-reviewed journals. We used the following wide range of search terms related to medication, drug, medicine, adherence, non-adherence, compliance, non-compliance, patient, experience, fear, beliefs, knowledge, attitudes, behaviour, communication, reason, and cause. Relating to the study design, our search terms were: qualitative, interview, focus group, questionnaire, observation, study, and research. An example of the search strategy is presented in the included appendix material (Appendix A).

### 2.2. Inclusion and Exclusion Criteria

We were interested in the phenomena leading to medication adherence and non-adherence from the patient perspective. Therefore, we included qualitative studies where the primary focus was understanding the complexity of medication adherence described by patients who were being treated for chronic conditions. We included original qualitative studies and systematic reviews if the study population consisted of people of 18 years or older and patients with at least one chronic condition. We also required that the primary focus was on patients’ experiences and attitudes towards medication adherence. We did not require comparison groups. It was mandatory that the researchers had used qualitative methods both for data collection and data analysis. We wanted to study the phenomena in general, so we excluded studies where the primary study population consisted of children or adolescents under 18 years or patients with an acute illness who were pregnant or drug or alcohol users. We also excluded conference papers, quantitative methods and mixed methods studies, as well as studies that collected data using qualitative methods, but data was analysed using quantitative methods.

### 2.3. Study Selection

The systematic searches for eligible articles retrieved 4404 studies. After duplicates were removed, the researchers (KK, AW, HL) independently screened the titles and abstracts for eligibility using the online software, Covidence. If one or two reviewers identified the article as relevant, we carried out a full-text review. We solved any disagreements via discussions and reaching a consensus. After the title and abstract screening, two reviewers (KK, AW) independently screened the full text of selected articles. Disagreements were resolved through discussions with the third reviewer (HL) for final inclusion. The articles were selected in several parts, which allowed the reviewers to have a regular discussion of the eligibility criteria, ensuring the same understanding of the criteria, and the criteria remaining the same throughout the article selection phase.

We did not assess the risk of bias of the included studies. As in many scoping reviews, the goal was to describe the phenomena surrounding patients and medication adherence [20].

### 2.4. Data Extraction

We constructed a template to carry out the data extraction using the Covidence online platform. Two reviewers (KK, HL) independently extracted the data, and the results were reviewed and verified by both reviewers for quality and clarity. We resolved the discrepancies by discussions and reaching a consensus. The data extraction template first focused on the study design, illness, context and concept of the studies, as well as barriers and facilitators to medication adherence. After extracting a third of the studies, we constructed a more specific classification for barriers and facilitators to medication adherence and re-extracted the material from the beginning with the wider list of items. We elaborated this classification further during the analysis of the results. We noted patients’ knowledge of their illness and its treatment. Motivation and behaviour skills seemed to be essential and correlated to good medication self-management during the analysis. Therefore, we decided to apply the IMB model as part of the classification of facilitators to medication adherence [15,16]. The authors provide by request the final list of data items documented in the Covidence extraction form.

## 3. Results

We included 89 original peer-reviewed articles in this scoping review (Figure 1). The study design in all the articles was qualitative and carried out in community or outpatient settings (Table A1). The studies were conducted in 36 different countries: The United States (*n* = 19), The United Kingdom (*n* = 10), South Africa (*n* = 4), Australia (*n* = 3), Canada (*n* = 3), Malaysia (*n* = 3), The Netherlands (*n* = 2), Sweden (*n* = 2), Indonesia (*n* = 2), Iran (*n* = 2) and one study from each of the following countries: Belgium, Norway, Portugal, Spain, Switzerland, Germany, Ireland, France, Italy, Singapore, New Zealand, Taiwan, Jordan, Pakistan, Kuwait, Saudi-Arabia, Vietnam, Uganda, Tanzania, Kenya, Eswatini, Ethiopia, Namibia and Lesotho. There was one study where both Nepal and Australia were involved and one study where Italy, Portugal and Poland were involved.

Our review covered 13 systematic reviews on medication adherence (Table A2). Seven of them focused on patients with cardiovascular disease or type two diabetes [21,22,23,24,25,26,27], one on patients with rheumatoid arthritis [28], one on patients with breast cancer [29], two on patients with chronic kidney disease or kidney transplants [10,30] and two with no specific illness [31,32].

There were 17 studies that had a behaviour theory-based approach to medication adherence (Table 1). The theories that appeared were: Andersen’s Behavioural Model [33,34], Roy Adaptation Model [35], Common-Sense Model of Self-Regulation [36], Social-Ecological Model [37,38], Therapeutic Alliance [39], Dowell’s Therapeutic Alliance Model and Leventhal’s Common Sense Model [40], Health Literacy Pathway Model [41], ABC Taxonomy and Three-Factor Model [32], Health Belief Model [42,43,44,45], Naturalistic Decision Model [46] and Stages of Change Model [47]. One of the studies did not have a theory-based approach in the beginning, but many of the findings fitted together with the Information–Motivation–Behaviour model [48].

The context of most of the studies was an outpatient setting, either in primary or secondary care (Table A1). The studies’ concept varied from the rationale of taking medication to understanding patients’ beliefs, practices, and reasons for nonadherence.

### 3.1. Barriers to Medication Adherence

Overall, the studies reported more barriers than facilitators to medication adherence. We inductively identified six subject areas with subcategories related to barriers to medication adherence based on the included qualitative studies (*n* = 89). The classification was data-driven, and we compiled it after extracting evidence from a third of the studies. We then went back to the beginning and re-extracted the data with the improved categorization. See Figure 2 and Figure 3.

#### 3.1.1. Patient-Specific Barriers

Patients may lack information or knowledge to understand their medication regimen properly. At the beginning of their disease, they may have received medication information and adherence counselling but without any follow-up, leading to patients being forgetful [49]. If the patient is extremely ill at the time of counselling, it may be challenging to adapt the information provided, and misunderstandings can occur. Patients can have poor awareness about the need to take medication as prescribed, and they tend to adjust their doses according to their understanding [46,50]. They may have incorrect or erroneous beliefs about medication [51]. They can lack motivation and think the disease is something they cannot control [52]. A lack of routines, being busy, or changes in practices for special occasions are risk points for medication adherence and can easily lead to missing doses or sleeping through dosing times [48].

Stress and helplessness can affect medication adherence [53]. Injectable drugs may feel unpleasant, and a patient may think injecting will destroy the body [52]. Patients’ physical disabilities can also be a barrier when administrating the medicine, which may require good eyesight or a steady hand [54]. Poor health literacy increases the adherence problem, and there can also be difficulties in understanding written language, especially if it is not written in a patient’s mother tongue [34,41]. Comorbidity may increase the probability of non-adherence [55].

#### 3.1.2. Illness-Specific Barriers

Contrary to healthcare professionals’ expectations, the disease is not always the priority for the patient [35,52,56]. It can be an unwanted episode, but not as important as other matters in life. A patient may have an adverse emotional reaction to the illness and judge life before the illness as more valuable. The required life changes may not be a priority. Patients may also rationalise that the disease is not so severe that they need to take their medication precisely as prescribed. Choosing to take or not to take medicines may depend on how seriously the patient assesses their situation [57].

Sometimes the challenge is that the patient has not accepted the illness or thinks it is someone else’s fault. The negative beliefs of illness or multiple diseases can increase barriers to medication adherence though it can differ from condition to condition [58]. Cancer can be understood as more life-threatening than diabetes, although diabetes can have grave consequences when not treated as required. The disease itself can cause fatigue and overwhelming tiredness, which negatively impact adherence [59].

#### 3.1.3. Medication-Specific Barriers

At the time of the onset of the illness, patients may lack the information on their condition or on the medication they need [60,61]. They can feel confused about the illness duration and prognosis [42,43]. Treatment can often seem time-consuming and complex to them [58,62]. Taking medication can be associated in patients’ minds with being sick, which can negatively influence adherence [35]. Difficulties in integrating medication into daily life can prevent patients from taking medication as prescribed. Working life may require shift work, and night shifts may make it difficult to have regular routines [63]. Besides, the illness may not have visible symptoms, and patients may not feel unwell [64]. Patients also fear that once they start a medication, this means they must continue taking it throughout their life [65].

If the medication information for a patient is inadequate and does not meet patients’ needs, they may use alternative information sources such as the internet [66]. A patient information leaflet in a medicine package may be difficult to understand. Warnings of side effects in the package sometimes make a patient decide not to take the medicine. Generic substitution may cause suspicions of the effect of a generic drug compared with the original product, thereby negatively affecting adherence [37]. Media can also influence opinions of the quality of drugs [67]. The desire of patients to self-regulate their lives may sometimes lead them to use non-prescription drugs instead of prescribed medicines [68].

Struggling with side effects seems to be a common barrier to medication adherence. Fear and the thought of not being safe with their medication may keep patients from taking it [60]. There are also physical barriers surrounding medication-taking: the size of the tablet can make it difficult to swallow, there can be unpleasant metallic after-taste or throat pain [59]. Needle phobia can prevent injecting insulin. A change from oral tablets to injectable drugs can be a drawback for patients [35].

#### 3.1.4. Healthcare and System-Specific Barriers

Poor access to healthcare and long waiting times cause poor medication adherence [55]. Fragmentation of treatment between multiple prescribers, a lack of communication between a general practitioner and a community pharmacist and poor coordination between primary and secondary care can lead to treatment problems. These, in turn, can lead to the discontinuation of care [55,69,70].

A lack of support and empathy from healthcare providers and a paternalistic manner can negatively impact adherence [14,43,55,71]. Poor patient-provider relationships lead to insufficient patient counselling and leave the patient alone struggling with medication problems [43]. Without trust-based patient-provider communication, patients cannot freely discuss side effects and other concerns related to their medication [39,72]. The inability of healthcare professionals to discuss adherence problems with patients and take their concerns and experiences seriously can impact the self-efficacy of patients [73,74]. A lack of trust in doctors and questioning their expertise may increase the burden of the illness and have an essential influence on a patient’s adherence behaviour [60].

#### 3.1.5. Social and Culture-Specific Barriers

A stigma is a common reason for nonadherence, especially with HIV/AIDS and with non-communicable diseases [71]. Patients may not want anybody to know about their illness. The fear of being stigmatized can be so intense that the patient prefers not to take their medication if there is a possibility that someone might be watching. It can be difficult to reconcile work and illness [74]. A lack of support from significant others can have a substantial impact on adherence and control of the illness [75,76].

Patients can prefer traditional alternatives or homeopathic remedies or methods instead of conventional medicine because these are more “natural” [45,72,77]. Patients can have a strong religious faith and prioritize religious rituals instead of taking medicines. Fasting during Ramadan and holy water can have a significant impact on medication management and may be the leading cause to adjust the medication to fit better with religious situations and routines [71]. Patients may stop the medication if they believe that praying can cure them [78].

#### 3.1.6. Logistical and Financial Barriers

Financial burdens and costs of medicines are significant barriers to medication adherence [79]. Unemployment and economic difficulties can affect the ability to buy medicines. If a patient does not have enough money to buy necessities such as food and clothing, medicines are unlikely to be a priority [80]. Difficulties travelling to the clinic, especially in developing countries, can hinder good medication self-management [42]. If insurance coverage is not comprehensive enough or there is no insurance, the cost of medicine can be unbearable [80]. A medicine shortage and the availability of medicines at the clinic or pharmacy, especially in developing countries, can become a significant problem for the continuity of care [60].

### 3.2. Facilitators to Medication Adherence

We identified five subject areas related to facilitators of medication adherence. Because medication taking is related to individual behaviour, we used the Information–Motivation–Behavioural Skills (IMB) Model as a starting point for the analysis [15,16]. However, as medication adherence is a complex entity in addition to human behaviour, we observed healthcare and system-specific factors and logistical and financial factors (Figure 4 and Figure 5).

#### 3.2.1. Informational, Motivational and Behavioural Factors

A good understanding of the illness and its treatment and how medicines promote the quality of life is essential for adherence [68]. The ability to integrate medications into daily life improves adherence in self-managing chronic conditions [79]. Low toxicity, mild adverse effects and an oral route of the administration seem to promote medication adherence [66,79]. There are different tools to assist with medicine taking, such as pillboxes, clock or mobile alarms, or taking medications during regular TV and radio programs [49,55].

The patient’s motivation is an essential facilitator. Motivation improves if the patient understands the necessity of the medication, and it contributes to positive health benefits [80]. Significant life events can have a positive effect on medication adherence. If a serious complication occurs, the importance of preventing complications and maintaining health is highlighted and may lead to a re-evaluation of the patient’s priorities [35,64,81]. The desire to return to “normal life” is a powerful facilitator to medication adherence [53].

The concerns related to illness may improve adherence and motivation to take medication as prescribed [79,82]. If patients have lived through the experience of their disease and its further negative impact on functional abilities, medication adherence may increase [82]. Knowing that interrupting or changing medications would result in the disease worsening can increase the desire to self-manage medication better [79]. The treatment goals must be realistic and achievable for the patient [52].

Support from family and friends and colleagues at work support adherence. It may require the disclosure of the illness, which can be scary for the patient [60,71]. Social acceptance helps the patient to cope with the illness.

Self-efficacy is an essential skill when coping with practical problems in daily life. If the patient takes ownership of self-managing the medication and knows how to adjust medicines if the disease worsens, the chances for better adherence are higher [60,83]. Feeling responsible and having a strong belief in the efficacy of medication promote self-empowerment and create a positive attitude towards the medication [59].

#### 3.2.2. Healthcare and System-Specific Facilitators

A trust-based, collaborative and respectful patient-provider relationship is crucial for medication adherence [55]. Good access to healthcare and enough time for discussions are necessary for patients [57]. Sometimes a desire to please healthcare providers or fearing them may also facilitate adherence [55]. Patients wish for confidential communication and an ongoing dialogue with health care professionals [84]. Support from healthcare providers and freely accessible care appear to increase adherence [65].

#### 3.2.3. Logistical and Financial Factors

Financial flexibility is necessary for medication adherence. The balance between revenue and expenditure of the household makes it possible to buy essential commodities such as food, clothes, and medicines without prioritising [80]. Additionally, having good insurance coverage guarantees secure finances in contrast to having no insurance at all.

### 3.3. Summary of Findings

We identified an extensive range of barriers and facilitators to medication adherence and the studies were more often concerned with barriers than facilitators. We classified the barriers as patient-specific, illness-specific, medication-related, healthcare and system-related, sociocultural and logistical and financial factors. The facilitators we identified were information and knowledge of the disease and medication, individual and social motivation, adherence behaviour skills, healthcare and system-specific factors and logistical and financial factors. Some of these factors can act as barriers and facilitators, such as healthcare and system-related factors and logistical and financial factors.

We identified similar factors to medication adherence in the previous systematic qualitative reviews (*n* = 13, Table A2) as in the qualitative studies described above (*n* = 76, Table A1). The previous systematic review findings confirm our own findings and the complexity of medication adherence as a phenomenon.

Some of the included studies had a theory-based approach to medication adherence (*n* = 17) (Table 1). Using different theories helped to understand and explain patients’ actions related to taking their medication (Table 1).

## 4. Discussion

### 4.1. Main Findings

To improve medication adherence, better communication and better information on the disease and its medication appeared to be the crucial concepts for patients in this scoping review. Our findings confirm that medication adherence is a complex phenomenon that is only partly understood. A wide range of factors seems to influence this either positively or negatively or in both ways. Regardless of the study concept, our findings were similar from study to study. Patients have many concerns about their illness, and it seems that they do not commonly have enough information to make knowledge-based decisions for self-managing their care. Patients want to discuss their problems and fears with a healthcare provider, but there is often not enough time for that in a short appointment.

According to this scoping review, the illness was not always a priority for the person. There can be many other matters in life that people prioritise more than their own optimal disease self-management. For better medication adherence, healthcare providers need to pay more attention to patients’ thoughts and concerns and have more time to listen to their experience in relation to the disease. Patients highly value trust-based relationships with healthcare providers.

This scoping review tracked many barriers that can hinder patients’ intention to adhere to their medication taking. The complexity of the matter may explain why many interventions to improve medication adherence are not successful [3]. If the intervention targets only some of the barriers, positive outcomes may be lacking, despite good intentions. The adherence to one medicine does not either automatically mean adherence to other medicines. Thus, adherence can differ from treatment to treatment or from disease to disease [68]. Patients may make their own priorities about the medications they use. This phenomenon should be further researched.

Our review of qualitative studies indicates that more attention should be paid to the patients’ fear of side effects. This can be a barrier that affects medication taking and can lead to skipping doses. With good knowledge and open and trust-based discussion with a healthcare provider, the patient need not begin to doubt their treatment. It is also good to discuss the patient’s values and religious values. A well-informed patient should know how to adjust medication to fit with religious requirements. The better the healthcare providers know the reality of their patients; living situations, the better they can support their patients to become empowered to self-manage a complex medication regimen.

Barriers can exist that the healthcare provider has not taken into consideration. The patient may have obstacles to self-manage taking their medication, for example, the difficulties of injecting medicines, remembering to take their medication on time when working, or the fear of stigmatisation. Additionally, financial obstacles can be difficult to reveal. Ideally, health care professionals should meet the patient without any preconceptions and in a trusted environment to discuss the barriers and concerns related to medication.

A theory-based approach may help to understand the patient’s actions and behaviours. However, a minority of the research we found had a theoretical approach, and the theories applied varied. Different behavioural theories, also adherence-specific ones, aim to explain chronically ill patient’s behaviour and give a reasonable explanation of why the patients act as they do. According to those theories, the patient’s action depends on their behaviour. This, in turn, depends on the patient’s beliefs or expected outcomes. According to our findings, an IMB model explains factors influencing adherence related to patient’s behaviour. However, external circumstances affect adherence, such as financial problems or poor access to care, which have to be considered. Based on our analyses, the different behavioural theories are good tools but do not fully explain complex adherence behaviour.

There is a need to generate new theory-based approaches to medication adherence since the current behavioural theories are not completely successful in explaining the complex phenomenon of adherence. There are also numerous adherence scales, which are very diverse and difficult to compare, so the research may need to be focused on comparing existing scales and determining which are most reliable. Qualitative research provides new insights into patient experiences and daily life struggles with their diseases and medicine taking to be incorporated in further development of the adherence measures and models. A good example of such a novel and promising conceptual approach is a model of medication-related burden and patients’ lived experiences with medicine, which builds on a meta-synthesis of qualitative studies [12].

Our review covered 13 systematic reviews, of which 11 were disease-specific, and 2 were generic. The illness-specific systematic reviews pointed out that patients often had misinterpretations of their illness, which prevented their adherence to medication. Clarifying these issues, having time and support, including from family members, were key recommendations to improve adherence in these reviews.

In our findings, there were more studies on barriers to medication adherence than facilitators. This is an interesting finding and can be affected by the fact that medication adherence is poorly understood, at least how it can be improved. When trying to understand the patient’s struggle with complex medication regimens, it may have caused the focus of research to go more towards barriers than facilitators. This scoping review may help to better understand the broader picture of adherence and to find interventions and strategies to improve it.

To our knowledge, this is the first scoping review on patient-related factors of medication adherence based on qualitative research. We conclude that well-informed patients and trustful patient-provider relationships are at the centre of improving medication adherence. Self-efficacy is crucial and empowers the patient to control and self-manage the disease and adjust the medication when necessary. Patient motivation needs to be monitored and supported. Moreover, patients need help integrating the medication regimen into their daily lives and to have routines. Support from significant others is essential too. They can support the patient in a life-long journey with the disease and give motivation for good medication self-management.

However, more research is needed to understand the patient’s reality. This scoping review clarifies the contributing factors of nonadherence and why the outcomes of interventions to improve adherence can be poor. The observations presented in this scoping review are useful when planning more effective interventions to increase medication adherence.

### 4.2. Strengths and Limitations

This scoping review of qualitative studies provides new information on people’s medicine-taking behaviour, which may not have been used to the best advantage. The strengths of this scoping review include an extensive literature search and review, followed by a thorough categorization of the barriers and facilitators of medication adherence. The literature searches were made with the support of an experienced librarian, and we had good coverage of qualitative studies where the primary focus was patients’ experiences and attitudes towards medication adherence. To avoid a selection bias, there were three researchers involved in the selection process. The data was thoroughly extracted and analysed to define the overarching categories.

A limitation is that since we focused on the qualitative aspects, we cannot conclude the magnitude of the effect of several factors influencing adherence. We also limited our search to studies in English, which may be a source of bias. The studies reported more barriers than facilitators, which may be another limitation. On the other hand, it describes the fact that barriers have been better recognized than facilitators. More research should be focused on the factors that have been able to help patients to commit to their disease and medication self-management. More research is also needed to elaborate on new theoretical models. This scoping review provides a good basis for building up more comprehensive theoretical models on medication adherence.

## 5. Conclusions

This scoping review highlighted a wide range of barriers and facilitators. The barriers seem to be better known than the facilitators. There is a need for better recognition of facilitators. We may need to increase the qualitative research of medication adherence to better understand the patients lived experiences that direct their medicine-taking behaviour. This information is needed to find new interventions and approaches to increase medication adherence, compare existing adherence scales, and build up more comprehensive theoretical models on medication adherence.

Patients wish to discuss their concerns about medications. Better communication and information appear to be among the most crucial factors for patients. The factors presented in this scoping review may help clinicians who communicate with patients having issues with adherence. The findings of this scoping review may also help those who plan further interventions to build up a more comprehensive approach to improve medication adherence.

## Figures and Tables

**Figure 1 pharmaceutics-13-01100-f001:**
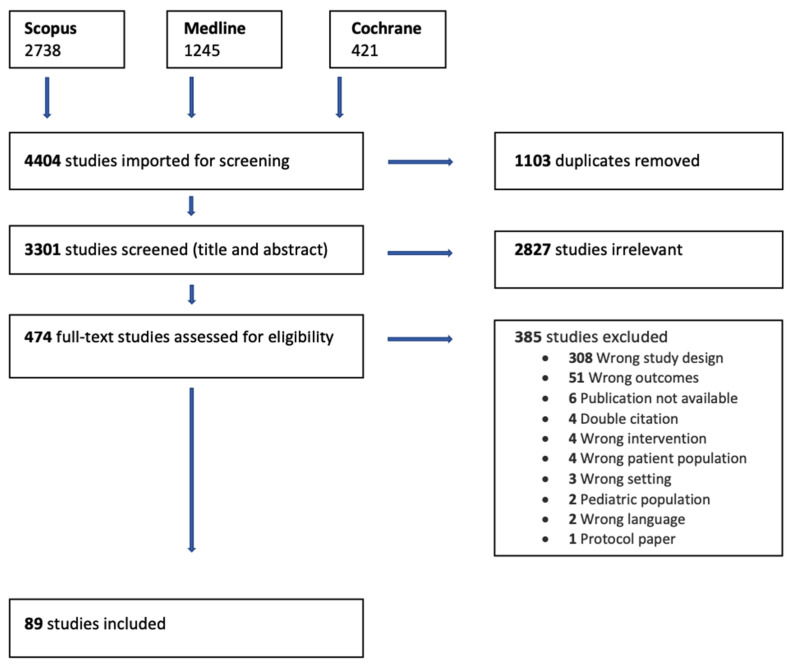
PRISMA flow diagram of the study selection process.

**Figure 2 pharmaceutics-13-01100-f002:**
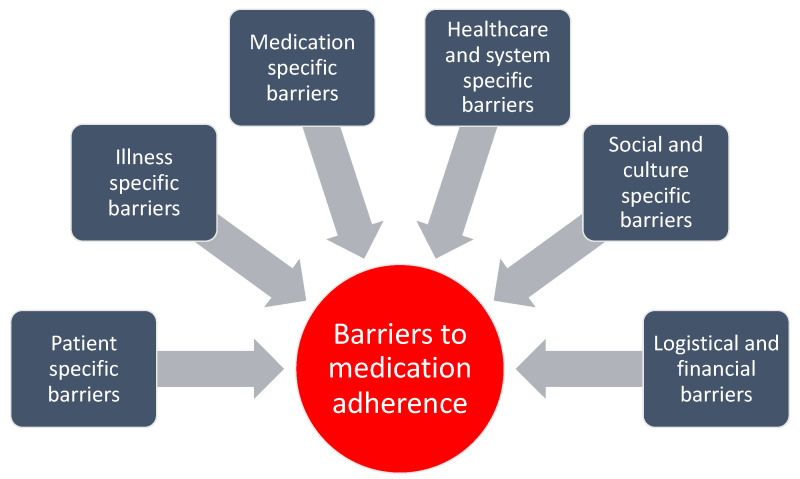
The identified key barriers to medication adherence based on the included qualitative studies (*n* = 89).

**Figure 3 pharmaceutics-13-01100-f003:**
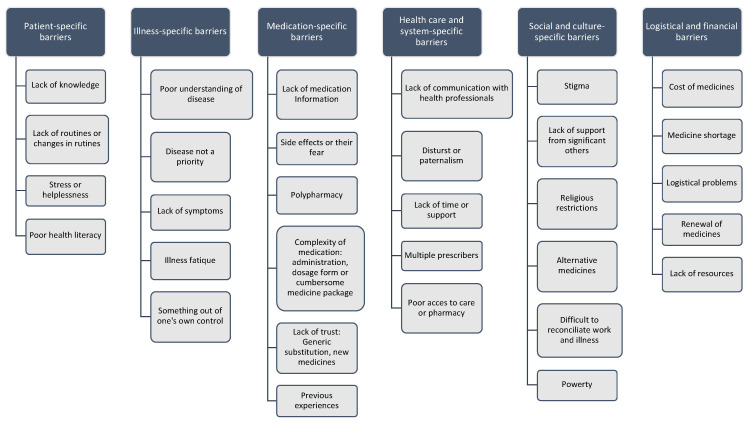
Subcategorisation of barriers to medication adherence arising from the included qualitative studies (*n* = 89).

**Figure 4 pharmaceutics-13-01100-f004:**
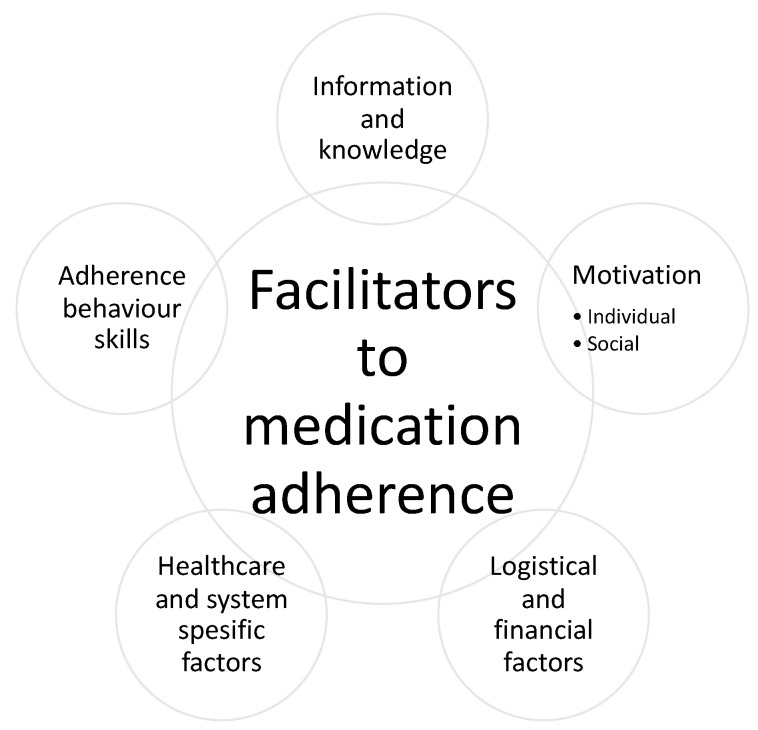
The identified key facilitators to medication adherence based on the included qualitative studies (*n* = 89).

**Figure 5 pharmaceutics-13-01100-f005:**
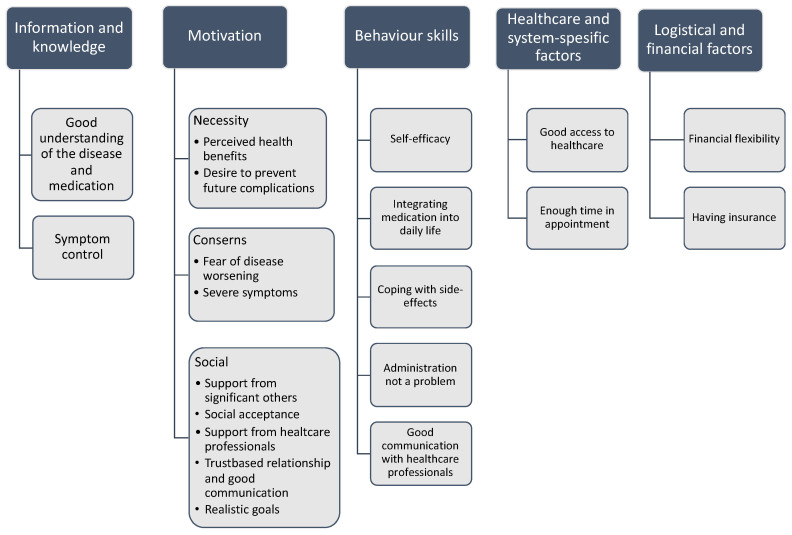
Categories and subcategories of facilitators to medication adherence arising from the included qualitative studies (*n* = 89). We used the Information–Motivation–Behaviour skills (IMB) model as part of the classification.

**Table 1 pharmaceutics-13-01100-t001:** Summary of the theories used in the included studies.

Theory	Medication Therapy	Study
**ABC Taxonomy and Three Factor Model**	Hypertension, heart disease, COPD, asthma	Maffoni et al., 2020
**Andersen’s Behavioural Model**	Antiretroviral therapy Antiretroviral therapy	Holtzman et al., 2015 Schatz et al., 2019
**Common-Sense Model of Self-Regulation (CSM)**	Glaucoma medication	McDonald et al., 2019
**Dowell’s Therapeutic Alliance**	Cardio-protective medication	Lambert-Kerzner et al., 2015
**Dowell’s Therapeutic Alliance Model and Leventhal’s Common-Sense Model**	Use of prescription medicines in general	Kucukarslan et al., 2012
**Health Belief Model**	Heart medication Clopidogrel Rheumatoid arthritis Hypertension	Garavalia et al., 2009 Garavalia et.al., 2011 Oshotse et.al., 2018 Rahmawati et al., 2018
**Health Literacy Pathway Model**	Diabetes type 2 medication	Huang et al., 2020
**Information-Motivation-Behaviour Skills (IMB) Model of Adherence**	Chronic hepatitis C therapy	Evon et al., 2015
**Naturalistic Decision-making Model**	Heart failure	Meraz et.al., 2020
**Roy Adaptation Model**	Diabetes type 2 medication	Bockwold et al., 2017
**Social Ecological Model**	Cardiovascular medication Antiretrovial therapy	Petterssen et al., 2018 Becker et al., 2020
**Stages of Change Model**	Anti-diabetic medication	Sapkota et al., 2018

## Data Availability

The available data is included in Appendix A, Appendix B and Appendix C. Other data used in this study are available on request from the corresponding author.

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
