# Peer review of "Factors Contributing to Medication Adherence in Patients with a Chronic Condition: A Scoping Review of Qualitative Research"

_pharmaceutics, 2021, doi:10.3390/pharmaceutics13071100_

Round 1

Reviewer 1 Report

The authors should be congratulated for an excellent and comprehensive review regarding factors contributing to medication adherence in patients with a chronic condition. Medical adherence is a complex phenomenon that has a great influence on clinical outcomes. Therefore, better understanding of patients' views on adherence is critical to successful long-term outpatient treatment. Nevertheless, I have one doubt regarding this manuscript. The Authors included articles that were published from January 2009 to September 2019, so they did not take into account publications from the last 20 months (counting until May 2021). In my opinion, this valuable manuscript should be updated prior to publication.

Author Response

Response to Reviewer 1 Comments

Thank you for the valuable comments concerning our manuscript “Factors Contributing to Medication Adherence in Patients with a Chronic Condition: a Scoping Review of Qualitative Research”. We have taken the comments into consideration and revised the manuscript accordingly.  Please find below a detailed description of our revisions as a response to the comments and recommendations.  

The referees’ comments are in italic, our responses in basic font (in red).

 We are hopeful that our revised manuscript will be of that scientific quality that it can be accepted for publication.

Point 1: The authors should be congratulated for an excellent and comprehensive review regarding factors contributing to medication adherence in patients with a chronic condition. Medical adherence is a complex phenomenon that has a great influence on clinical outcomes. Therefore, better understanding of patients' views on adherence is critical to successful long-term outpatient treatment. Nevertheless, I have one doubt regarding this manuscript. The Authors included articles that were published from January 2009 to September 2019, so they did not take into account publications from the last 20 months (counting until May 2021). In my opinion, this valuable manuscript should be updated prior to publication.

Response: Thank you for this valuable comment. We have updated the search from September 2019 to June 9th, 2021, as suggested (lines 91-103; Figure 1 lines 193-196; Table 1 lines 233-234; Table A2; Table A3).

Reviewer 2 Report

Overall, I believe this is a very good work. It employs an interesting approach to the study of medication adherence, which, as the authors explore, are particularly relevant today. 

Nevertheless, I believe there are important changes to be made, detailed in the annexed file.

Author Response

Thank you for all the valuable comments concerning our manuscript “Factors Contributing to Medication Adherence in Patients with a Chronic Condition: a Scoping Review of Qualitative Research”. We have taken all the comments into consideration and revised the manuscript accordingly. Please see the attachment below of a detailed description of our revisions as a response to the comments and recommendations.  

Round 2

Reviewer 2 Report

Thank you for taking the time to answer my recommendations in detail. I believe you have written a very interesting article and would like to congratulate you for it.

Kind regards.